# Human Sox4 facilitates the development of CXCL13-producing helper T cells in inflammatory environments

Hiroyuki Yoshitomi[1,2], Shio Kobayashi[3], Aya Miyagawa-Hayashino[4], Akinori Okahata[5], Kohei Doi[5], Kohei Nishitani[6], Koichi Murata[6], Hiromu Ito[5], Tatsuaki Tsuruyama[7], Hironori Haga[4], Shuichi Matsuda[5] & Junya Toguchida[1,2]

In human inflammatory sites, PD-1$^{hi}$CXCR5$^-$CD4$^+$ T cells are involved in the formation of ectopic lymphoid-like structures (ELSs) by the secretion of chemokine CXCL13, but how the transcription of CXCL13 is regulated in CD4$^+$ T cells is still unclear. Here we show that Sox4 is a key transcription factor for CXCL13 production in human CD4$^+$ T cells under inflammatory conditions. In vitro TGF-β$^+$, IL-2-neutralizing culture conditions give rise to PD-1$^{hi}$CXCR5$^-$CD4$^+$ T cells that preferentially express CXCL13, and transcriptome analysis and lentiviral overexpression indicate Sox4 association with the CXCL13 transcription. In vivo, Sox4 is significantly upregulated in synovial CD4$^+$ T cells, when compared with blood CD4$^+$ T cells, from patients with rheumatoid arthritis (RA), and further correlates with ELS formation in RA synovium. Overall, our studies suggest that Sox4 contributes to CXCL13 production and ELS formation at inflammatory sites in humans.

[1] Department of Regeneration Science and Engineering, Institute for Frontier Life and Medical Sciences, Kyoto University, 53 Kawahara-cho, Shogoin, Sakyo-ku, Kyoto 606-8507, Japan. [2] Department of Cell Growth and Differentiation, Center for iPS Cell Research and Application, Kyoto University, 53 Kawahara-cho, Shogoin, Sakyo-ku, Kyoto 606-8507, Japan. [3] Joslin Diabetes Center, 1 Joslin Pl, Boston, MA 02215, USA. [4] Department of Diagnostic Pathology, Kyoto University Hospital, 54 Kawahara-cho, Shogoin, Sakyo-ku, Kyoto 606-8507, Japan. [5] Department of Orthopaedic Surgery, Kyoto University Graduate School of Medicine, 54 Kawahara-cho, Shogoin, Sakyo-ku, Kyoto 606-8507, Japan. [6] Department of Advanced Medicine for Rheumatic Diseases, Kyoto University Graduate School of Medicine, 54 Kawahara-cho, Shogoin, Sakyo-ku, Kyoto 606-8507, Japan. [7] Department of Drug Discovery Medicine, Kyoto University Graduate School of Medicine, Yoshida-Konoe-Cho, Sakyo-Ku, Kyoto 606-8501, Japan. Correspondence and requests for materials should be addressed to H.Y. (email: yositomi@kuhp.kyoto-u.ac.jp)

One feature of human local inflammatory sites is that CXCL13-producing PD-1$^{hi}$CXCR5$^−$CD4$^+$ T cells contribute to the formation of ectopic (or tertiary) lymphoid-like structures (ELSs)[1–7]. These ELSs support immune responses related to infection, correlate with better prognosis in cancers, and stimulate autoantibody production in autoimmune diseases[3,6–10]. In secondary lymphoid organs such as the lymph nodes and tonsils, contrary to ELSs, human PD-1$^{hi}$CXCR5$^+$ follicular helper T (Tfh) cells, which mediate class switching and the affinity maturation of antibodies in germinal centers (GCs) through the activity of the master transcription factor BCL6[11–13], secrete CXCL13[14,15]. Although local PD-1$^{hi}$CXCR5$^−$CD4$^+$ T cells that express CXCL13 and interleukin (IL)-21 at the inflamed sites are referred to as Tfh-like cells[7], these cells do not show elevated BCL6 expression[2,4,5]. Thus, the transcriptional regulation that mediates CXCL13 production by PD-1$^{hi}$CXCR5$^−$CD4$^+$ T cells at the inflammatory sites remains to be explained.

A recent analysis of CD4$^+$ T cells of patients with rheumatoid arthritis (RA) using mass cytometry and transcriptomics revealed a population of PD-1$^{hi}$CXCR5$^−$CD4$^+$ T cells that is a distinct CD4$^+$ T-cell subset, expands in the blood of RA patients, and contributes to RA pathogenesis[5]. In addition to their B-helper activities, PD-1$^{hi}$CXCR5$^−$CD4$^+$ T cells, especially in locally inflamed joints, exert a heightened ability to produce CXCL13 compared with blood cells[2,5]. Consistent with this, transforming growth factor (TGF)-β simulation and a limited availability of IL-2 (IL-2-limiting) have been shown to have crucial roles in the in vitro differentiation of CXCL13-producing human CD4$^+$ T cells[16]. These findings collectively imply that local inflammatory conditions could be involved in the development of CXCL13-producing PD-1$^{hi}$CXCR5$^−$CD4$^+$ T cells, likely by regulating the expression of transcription factors.

In this study, we explore transcription factors related to CXCL13-producing CD4$^+$ T cells at local inflammatory sites. For this purpose, we differentiate CXCL13-producing PD-1$^{hi}$CXCR5$^−$CD4$^+$ T cells under inflammatory conditions in vitro and conduct transcriptome analysis. Sox4 is the only transcription factor that fulfills the screening criteria; in RA, it is upregulated in vitro, in a TGF-β-positive and IL-2-limiting condition, and in CD4$^+$ T cells in local inflammatory sites compared with blood CD4$^+$ T cells. Furthermore, lentiviral transduction of the Sox4 gene in human naive CD4$^+$ T cells induces an intense production of CXCL13, and Sox4 expression in RA synovium is significantly associated with ELS formation. These data collectively indicate that Sox4 expression in human CD4$^+$ T cells contributes to the mechanisms of chronic inflammation via CXCL13-dependent ELS formation at local inflammatory sites.

## Results

### Induction of CXCL13-producing PD-1$^{hi}$CXCR5$^−$CD4$^+$ T cells.
To investigate the association between the inflammatory environment and PD-1$^{hi}$CXCR5$^−$CD4$^+$ T cells, we differentiated healthy human naive CD4$^+$ T cells under several inflammatory conditions in vitro. TGF-β-positive conditions induced CXCL13-producing CD4$^+$ T cells that were highly positive for PD-1 and negative for CXCR5, whereas Th1- and Th2-polarizing conditions or a combination of proinflammatory cytokines alone did not induce CXCL13 or PD-1 (Fig. 1a, b). An activation marker, human leukocyte antigen (HLA) Class II, which is a hallmark of PD-1$^{hi}$CXCR5$^−$CD4$^+$ T cells[5], was preferentially expressed by PD-1$^{hi}$ cells differentiated in TGF-β-positive conditions, but by PD-1$^−$ cells under TGF-β-negative conditions (Fig. 1c). In some inflammatory diseases, IL-2 levels at the local inflammatory sites are limited because of low IL-2 production by resident or infiltrating cells[17] and IL-2 consumption by regulatory T (Treg) or

dendritic cells[4,18,19]. To investigate whether the limited availability of IL-2 affected CXCL13-producing PD-1$^{hi}$CXCR5$^−$CD4$^+$ T cells, we added IL-2-neutralizing antibody to the inflammatory environment, which resulted in a significant upregulation of CXCL13 production by PD-1$^{hi}$CXCR5$^−$CD4$^+$ T cells (Fig. 1a, b and Supplementary Fig. 1, 2). Specifically, TGF-β-positive, IL-2-limiting conditions, which are consistent with local inflamed sites in several inflammatory diseases[2,4,16,17], gave rise to CXCL13-producing PD-1$^{hi}$CXCR5$^−$CD4$^+$ T cells in vitro.

### Transcriptome analysis of CXCL13-producing CD4$^+$ T cells.
To address the transcription factors related to CXCL13-producing PD-1$^{hi}$CXCR5$^−$CD4$^+$ T cells at inflammatory sites, we enriched CXCL13-producing human CD4$^+$ T cells from in vitro-differentiated blood CD4$^+$ T cells and conducted a transcriptome analysis. The addition of proinflammatory cytokines to the TGF-β-positive, IL-2-limiting condition significantly enhanced cell proliferation and CXCL13 induction, and tracking cell division showed that CXCL13-producing cells were enriched in divided cells (Fig. 2a and Supplementary Fig. 3). By sorting cells with more than one division during the differentiation in the presence of TGF-β plus IL-1β, TGF-β plus IL-6, or IL-12 alone, we could obtain 55%, 69%, and 1.4% CXCL13-positive cells, respectively (Fig. 2b). Transcriptome analysis showed that signature genes for PD-1$^{hi}$CXCR5$^−$CD4$^+$ T cells of RA, such as CXCL13, SH2D1A, CCR2, TIGIT, MAF, and TOX[5], were upregulated in the TGF-β plus IL-1β and TGF-β plus IL-6 groups (Supplementary Table 1). Eighty-five probes showed more than fourfold increased expression of both TGF-β-positive groups compared with IL-12-treated cells or with naive CD4$^+$ T cells. To rule out genes related to Treg cells that might be induced by TGF-β[20], 19 probes that were upregulated fourfold in human blood Treg[21] were excluded (Fig. 2b). In the remaining 66 probes (Supplementary Table 2), 3 genes, CLIC3 (chloride intracellular channel protein 3), NLK (nemo like kinase), and SOX4 (SRY-box 4), were localized in the nucleus, but SOX4 was the only transcription factor that fulfilled the screening criteria.

### Regulation of Sox4 expression in human CD4$^+$ T cells.
Sox4, a member of the Sry-related high-mobility group (HMG) box (Sox) family, regulates T-cell differentiation in the thymus and population expansion of pro-B cells[22,23], and is a downstream target of the TGF-β signaling pathway[24,25]. Naive human CD4$^+$ T cells expressed Sox4 slightly and stimulation with TGF-β for 24 h significantly upregulated their Sox4 expression (Fig. 2c, d). Under TCR stimulation, the combination of TGF-β and proinflammatory cytokines upregulated SOX4 expression, as did TGF-β alone (Fig. 2e). As previously reported in other cell types[24,25], inhibitors of TGF-β signaling or of Smad3, a molecule downstream of TGF-β, attenuated the TGF-β-induced SOX4 upregulation in human naive CD4$^+$ T cells (Fig. 2f). We further investigated the effects of IL2-limiting on Sox4 induction by using a neutralizing anti-IL-2 antibody, which significantly upregulated both Sox4 and CXCL13 expression in cells differentiated for 3 days (Fig. 2g, h). Collectively, these findings indicate that TGF-β-positive, IL-2-limiting inflammatory conditions contribute to Sox4 expression in human CD4$^+$ T cells.

### Sox4 facilitates CXCL13 production in human CD4+ T cells.
To investigate whether Sox4 is functionally involved in CXCL13 production by CD4$^+$ T cells, the Sox4 gene was transduced into human naive CD4$^+$ T cells with yellow fluorescent protein (YFP)-expressing lentiviral vectors (Supplementary Fig. 4). Flow cytometry analysis showed that YFP-positive Sox4-expressing CD4$^+$ T cells highly expressed CXCL13 compared with

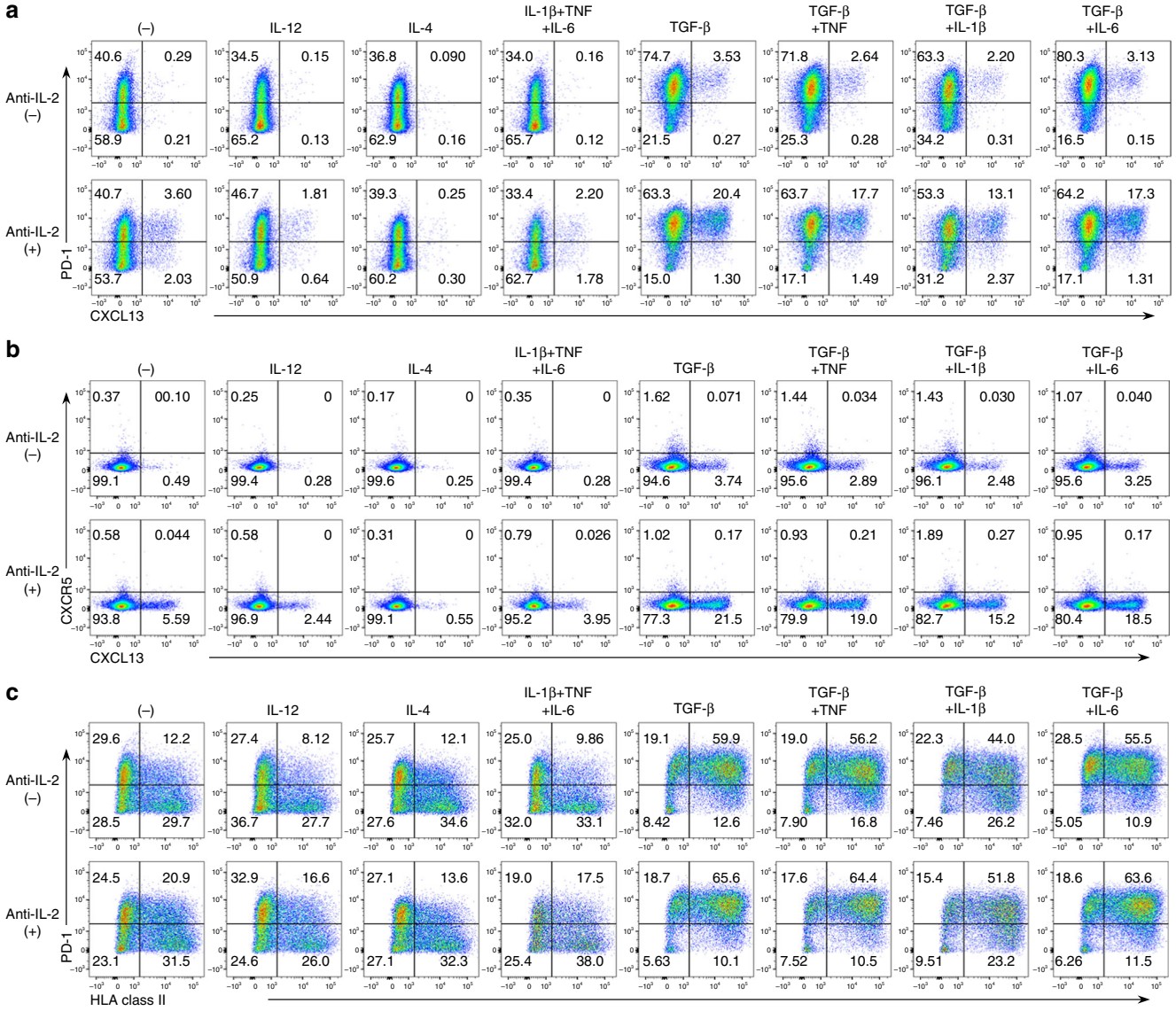

**Fig. 1** TGF-β-positive, IL-2-limiting conditions give rise to CXCL13-producing PD-1$^{hi}$CXCR5$^{-}$CD4$^{+}$ T cells in vitro. **a–c** Healthy human naive CD4$^{+}$ T cells were differentiated with TCR stimulation and the indicated cytokines in the presence or absence of neutralizing anti-IL-2 antibody for 5 days. Representative dot plots of PD-1 and intracellular CXCL13 (**a**), CXCR5 and CXCL13 (**b**), and PD-1 and HLA class II (**c**) are shown

mock-transduced CD4$^{+}$ T cells (Fig. 3a). The expression of *CXCL13* messenger RNA in sorted Sox4-transduced CD4$^{+}$ cells was about 35 times higher than that in sorted mock-transduced cells (Fig. 3b). Furthermore, lentiviral knockdown of Sox4 in naive CD4$^{+}$ T cells significantly downregulated CXCL13 induction (Supplementary Fig. 5). These findings indicate the physiological significance of Sox4 in CXCL13 production by human CD4$^{+}$ T cells under inflammatory conditions.

We further investigated whether CXCL13 production by Sox4-transduced CD4$^{+}$ cells was affected by the surrounding environment, because TGF-β is known to affect both the expression and activity of transcription factors; e.g., TGF-β induces RORγt expression but inhibits RORγt production of IL-17[26]. Without the addition of TGF-β, about 10% of Sox4-transduced CD4$^{+}$ cells produced CXCL13. The addition of TGF-β significantly upregulated dose-dependent CXCL13 production (Fig. 3c). Similarly, about 60% of Sox4-transduced CD4$^{+}$ cells were positive for CXCL13 under IL-2-limiting (Fig. 3d). These findings, together with the results about the regulation of Sox4 expression (Fig. 2c–h), collectively indicate that the function of

Sox4 could be affected by the status of the intracellular signals and that TGF-β stimulation and low IL-2 levels promote both Sox4 expression and the subsequent CXCL13 production.

**Analysis of domain functions in SOX4/CXCL13 axis**. Sox4 comprised an N-terminus domain, HMG domain, glycine-rich region (GRR), serine-rich region (SRR), and transactivator domain (TAD) at the C terminus (Fig. 4a). The N-terminus domain, HMG domain, which binds to DNA, and TAD, which facilitates Sox4 transcription activity by binding to other transcription factors[27], are highly conserved. To investigate the domain functions crucial for the Sox4/CXCL13 axis in human CD4$^{+}$ T cells, we truncated each domain of human Sox4 and transduced them by lentivirus into human naive CD4$^{+}$ T cells. Consistently, neither HMG- nor TAD-truncated Sox4 induced CXCL13 production, but truncations of the N terminus, GRR, or SRR did (Fig. 4b), indicating that the transcription activity of Sox4 is crucial for regulating CXCL13 production in human CD4$^{+}$ T cells. The N-terminus domain, HMG domain, and TAD of Sox4 are almost completely conserved in humans and mice

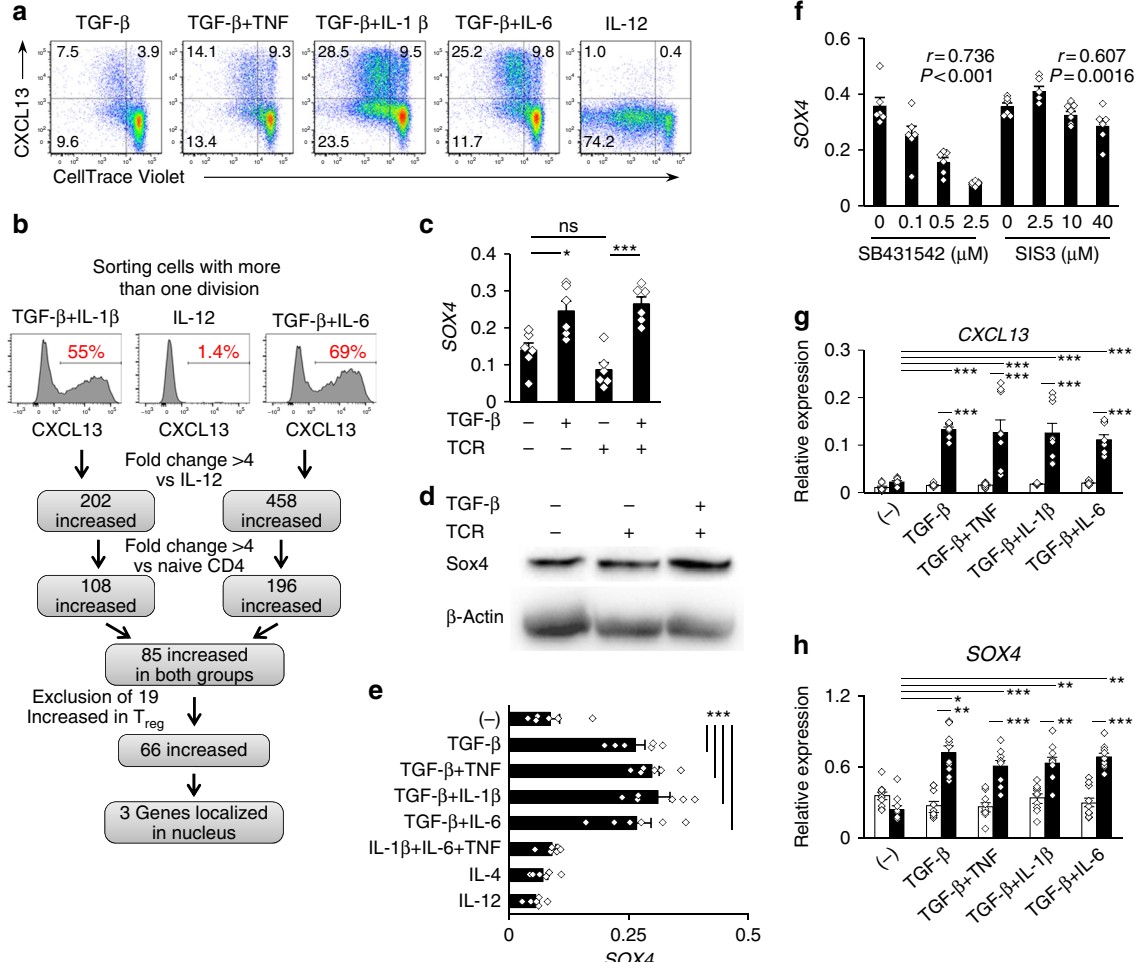

**Fig. 2** Transcriptome analysis identified Sox4 as a transcription factor relating to CXCL13-producing CD4$^+$ T cells. **a** Human blood CD4$^+$ T cells labeled with CellTrace™ Violet were differentiated in the presence of the indicated cytokines and neutralizing anti-IL-2 antibody for 5 days. Quadrants define CXCL13-positive/negative cells, and cells with 0–1 divisions and cells with $\geq 2$ divisions. **b** Outline for the screening of candidate transcription factors. **c**, **d** Sox4 expression in human naive CD4$^+$ T cells unstimulated or stimulated with anti-CD3/CD28 antibodies for 24 h in the presence of TGF-β was assessed by quantitative PCR (**c**) or immunoblotting (**d**). **e**, **f** Human naive CD4$^+$ T cells were cultured with TCR stimulation in the presence of the indicated cytokines (**e**) or TGF-β plus a TGF signal inhibitor, SB431542, or a SMAD3 inhibitor, SIS3 (**f**) for 24 h. Relative SOX4 expression was measured by quantitative PCR. **g**, **h** Relative mRNA expression of CXCL13 (**g**) and Sox4 (**h**) assessed by quantitative PCR in naive human CD4$^+$ T cells differentiated with TCR stimulation and the indicated cytokines in the presence (solid) or absence (open) of neutralizing anti-IL-2 antibody for 3 days. Cumulative data ($n = 6$ from two experiments in **c**–**e**, and **f**; $n = 9$ from three experiments in **g**, **h**) are presented as corresponding dots and the mean ± SEM. *$P < 0.05$, **$P < 0.01$, and ***$P < 0.001$ by one-way ANOVA, Tukey's test, and $r$ and $P$-values by Pearson's correlation analysis. Numbers in flow cytometry plots indicate the percentage of cells in the indicated region

(Supplementary Fig. 6). As expected, overexpression of mouse Sox4 or human Sox4 in human CD4$^+$ T cells significantly induced CXCL13 production (Fig. 4c). However, mouse CD4$^+$ T cells do not secrete CXCL13[28]. As such, we investigated whether the Sox4/CXCL13 axis is present in mouse CD4$^+$ T cells. Indeed, the upregulation of Sox4 in mouse CD4$^+$ T cells by TGF-β signaling or lentiviral transduction of either mouse or human Sox4 failed to induce Cxcl13 expression (Fig. 4d) despite the abundant expression of Sox4 protein (Supplementary Fig. 7). This finding is presumably because of differences in the transcription co-factors or promoter/enhancer regions between human and mouse CD4$^+$ T cells. Thus, the Sox4-mediated CXCL13 production by CD4$^+$ T cells might be an immunological function dependent on the species.

**Contribution of human Sox4 to Th1 or Th2 differentiation**. In mice, Sox4 has been reported to downregulate Th2 differentiation

by competing with GATA3 in DNA binding[24]. To investigate the involvement of Sox4 in human Th2 cell differentiation, we lentivirally transduced human Sox4 into human naive CD4$^+$ T cells and differentiated the cells under a Th2-polarizing condition. The overexpression of Sox4 had no effect on the differentiation of IL-4-producing Th2 cells in vitro (Fig. 4e). We also investigated the involvement of Sox4 in Th1 cell differentiation and found that the transduction of Sox4 significantly upregulated interferon (IFN)-γ-producing Th1 cells under a Th1-polarizing condition (Fig. 4f). Interestingly, CXCL13 was slightly expressed by Sox4-transduced CD4$^+$ T cells under a Th1-polarizing condition, but not under a Th2-polarizing condition (Fig. 4e, f). These findings imply a possible contribution of Sox4 on Th1 activity and CXCL13-dependent ELS formation in Th1-dominant inflammatory conditions. They further indicate that human Sox4 slightly promotes Th1 differentiation, but does not have an obvious effect on Th2 differentiation.

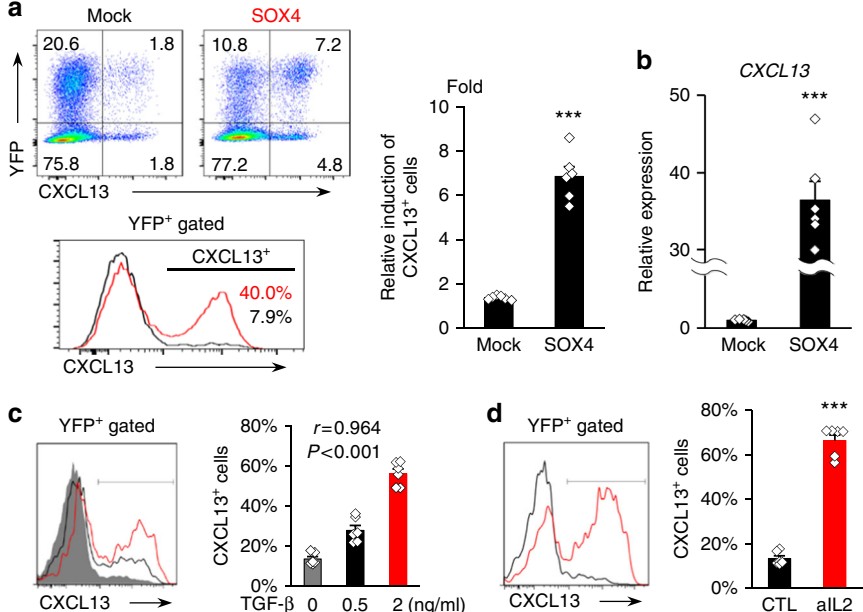

**Fig. 3** Sox4 is functionally involved in CXCL13 production by CD4[+] T cells. **a**, **b** Human naive CD4[+] T cells were transduced with mock or Sox4 by YFP-expressing lentivirus and differentiated with 2 ng/ml TGF-β. **a** Representative dot plots of YFP and intracellular CXCL13, representative histograms of CXCL13 in YFP[+] gated cells transduced with mock (black) or Sox4 (red), and graphical summary for the relative induction of CXCL13-positive cells in YFP[+] cells compared with YFP[−] cells are shown. **b** Relative expression of *CXCL13* in sorted mock- or Sox4-transduced naive CD4[+] T cells determined by quantitative PCR. **c**, **d** Sox4-transduced human naive CD4[+] T cells were cultured with TCR stimulation in the presence of TGF-β at 0 (gray), 0.5 (black), or 2 ng/ml (red; **c**), or in the presence (red) or absence (black) of neutralizing anti-IL-2 antibody (**d**). Representative histograms and a graphical summaries are shown. Cumulative data (*n* = 6 from two experiments) are presented as corresponding dots and the mean ± SEM. ****P* < 0.001 unpaired *t*-test, and *r* and *P*-values by Pearson's correlation analysis

**Transcription factors in CD4[+] T cells at inflammatory sites**. We have shown above that Sox4 upregulation is associated with the development of CXCL13-producing CD4[+] T cells in in vitro inflammatory conditions. To address whether Sox4 expression in CD4[+] T cells is also upregulated at local inflammation sites in human chronic inflammatory diseases, we investigated mRNA expression in the synovial and blood CD4[+] T cells of RA patients. *CXCL13* expression was significantly upregulated in RA synovial CD4[+] T cells compared with blood CD4[+] T cells (Fig. 5a), consistent with a previous report[2]. Also consistent with our in vitro results (Fig. 2c–h) is the significant upregulation of Sox4 expression in RA synovial CD4[+] T cells (Fig. 5a). We further investigated the expression of MAF, TOX, and Blimp1, because the upregulation of these transcription factors has been reported in the PD-1[hi]CXCR5[−]CD4[+] T cells of RA patients[5]. MAF, a transcription activator, and Blimp1, a transcription repressor, induces and suppresses the Tfh cell signature, respectively[29,30], and TOX, an HMG box protein, is required for CD4[+] T-cell development in the thymus[31]. The expressions of *MAF* and *TOX* were significantly upregulated in RA synovial CD4[+] T cells (Fig. 5a), and synovial PD-1[hi]CD4[+] T cells preferentially expressed MAF and TOX (Supplementary Fig. 8). On the other hand, *Blimp1* expression was not significantly different between synovial and blood CD4[+] T cells. These findings imply that the expression of MAF and TOX might also be regulated by the inflammatory environment in a manner similar to Sox4. To address this possibility, we investigated the expression of these transcription factors in CD4[+] T cells differentiated in vitro under inflammatory conditions. In IL-2-limiting, *MAF* expression was significantly upregulated in a TGF-β-positive condition. Interestingly, IL-2-limiting downregulated *MAF* expression, but upregulated *Sox4* expression (Figs. 5b, 2h). *Blimp1* expression was downregulated by both IL-2 neutralization and TGF-β stimulation (Fig. 5b). *TOX* expression was significantly upregulated in some IL-2-limiting, TGF-β-positive conditions

(Fig. 5b). These data collectively imply that factors related to inflammatory conditions such as TGF-β stimulation or IL-2 levels regulate the expression of Sox4, MAF, and TOX in vitro and probably also in inflammatory diseases.

**Regulation of PD-1[hi]CXCR5[−]CD4[+] T-cell signature genes**. To investigate the contribution of MAF, TOX, and Blimp1, in addition to Sox4, to the phenotype of CD4[+] T cells, we lentivirally transduced these transcription factors into human naive CD4[+] T cells. MAF and TOX slightly contributed to the induction of CXCL13 mRNA and protein, but far less so than did Sox4 (Fig. 5c, d). Blimp1 downregulated CXCL13 production (Fig. 5c, d), implying that the downregulated Blimp1 expression in inflammatory conditions (Fig. 5b) might also contribute to CXCL13 production.

CXCL13-producing PD-1[hi]CXCR5[−]CD4[+] T cells express certain Tfh signature genes, including PD-1, SH2D1A, IL-21, BATF, ICOS, TIGIT, CTSB, CD200, and SLAMF6[2–5]. Thus, we investigated the contributions of Sox4, MAF, TOX, and Blimp1 on these gene expressions (Fig. 5e). PD-1 *(PDCD1)* expression was upregulated by the transduction of MAF, TOX, and Sox4, but not of Blimp1. Sox4 also upregulated the expression of *TIGIT*, *CTSB*, *CD200*, and *SLAMF6*. Interestingly, *CXCR5* expression was downregulated by Sox4, implying a contribution of Sox4 to the negative expression of CXCR5 in CXCL13-producing CD4[+] T cells at inflammatory sites. These findings imply that multiple transcription factors might coordinately contribute to forming the phenotype of PD-1[hi]CXCR5[−]CD4[+] T cells in inflammatory conditions. Although Sox4 seemed not to contribute to the expression of *SH2D1A*, *IL21*, *BATF*, or *ICOS* in CD4[+] T cells, MAF and TOX did (Fig. 5e). Of note, MAF contributed to the expression of genes related to B-helper activities, such as *IL21*, *SH2D1A*, and *SLAMF6* (Fig. 5e). Collectively, these results imply

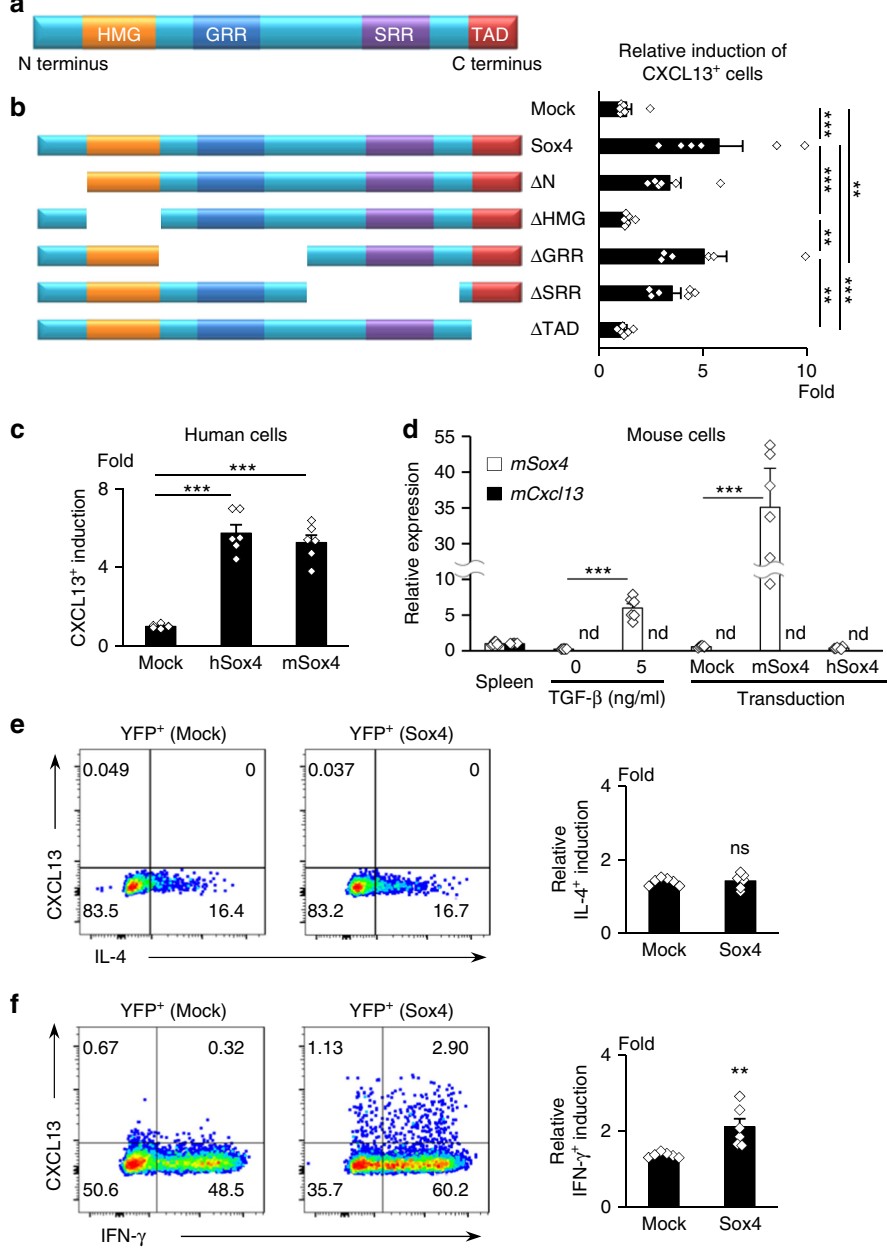

**Fig. 4** Conserved domains HMG and TAD are functionally crucial for CXCL13 induction in human CD4+ T cells. **a** Structure of Sox4 protein. **b** Wild-type Sox4 or Sox4 truncated at the indicated domain (left) were transduced into human naive CD4+ T cells, followed by differentiation for 5 days. Relative induction of CXCL13-positive cells in YFP+ cells compared with YFP− cells is shown (right). **c** Relative CXCL13 induction in differentiated human naive CD4+ T cells transduced with human and mouse Sox4. **d** Relative expression of *mSox4* and *mCxcl13* in mouse spleen tissue (defined as 1.0), in mouse CD4+ T cells cultured for 5 days with TCR stimulation and TGF-β, and in Sox4-transduced mouse CD4+ T cells differentiated for 5 days. **e**, **f** Mock- or Sox4-transduced human naive CD4+ T cells were differentiated under Th2-polarizing (**e**) or Th1-polarizing (**f**) conditions. Representative dot plots of YFP+ gated cells and relative IL-4 (**e**) or IFN-γ (**f**) induction in YFP+ cells compared with YFP− cells are shown. Cumulative data (n = 6) are presented as corresponding dots and the mean ± SEM. **P < 0.01 and ***P < 0.001 by one-way ANOVA, Tukey's test in **b** and **c**, and by unpaired t-test in **d**, **e**, and **f**; nd, not detectable; ns, not significant

that inflammatory conditions regulate the expression of Sox4, MAF, and TOX, and that the ELS formation regulated by CXCL13 expression depends on upregulated Sox4 expression in human CD4+ T cells.

**Sox4 expression correlates with ELS formation in RA synovium**. Finally, we investigated the expression and distribution of Sox4 in RA synovium. Sox4 expression was localized in cells infiltrating the sublining layer of RA synovium (Fig. 5f and Supplementary Fig. 9). We semi-quantitatively evaluated the

formation of ELSs and the Sox4 expression in areas with or without ELSs in RA synovium. Interestingly, ELS formation and Sox4 expression in RA synovium were significantly correlated (Fig. 5g). To further investigate the expression of Sox4 and CXCL13 in T cells infiltrating the RA synovium, we performed triple immunostainings of RA synovium with anti-CD3, Sox4, and CXCL13 antibodies. Consistent with the in vitro and in vivo findings reported above, T cells infiltrating the RA synovium preferentially expressed Sox4 and CXCL13 (Fig. 5h and Supplementary Fig. 10). Considering that PD-1hiCXCR5−CD4+ T cells

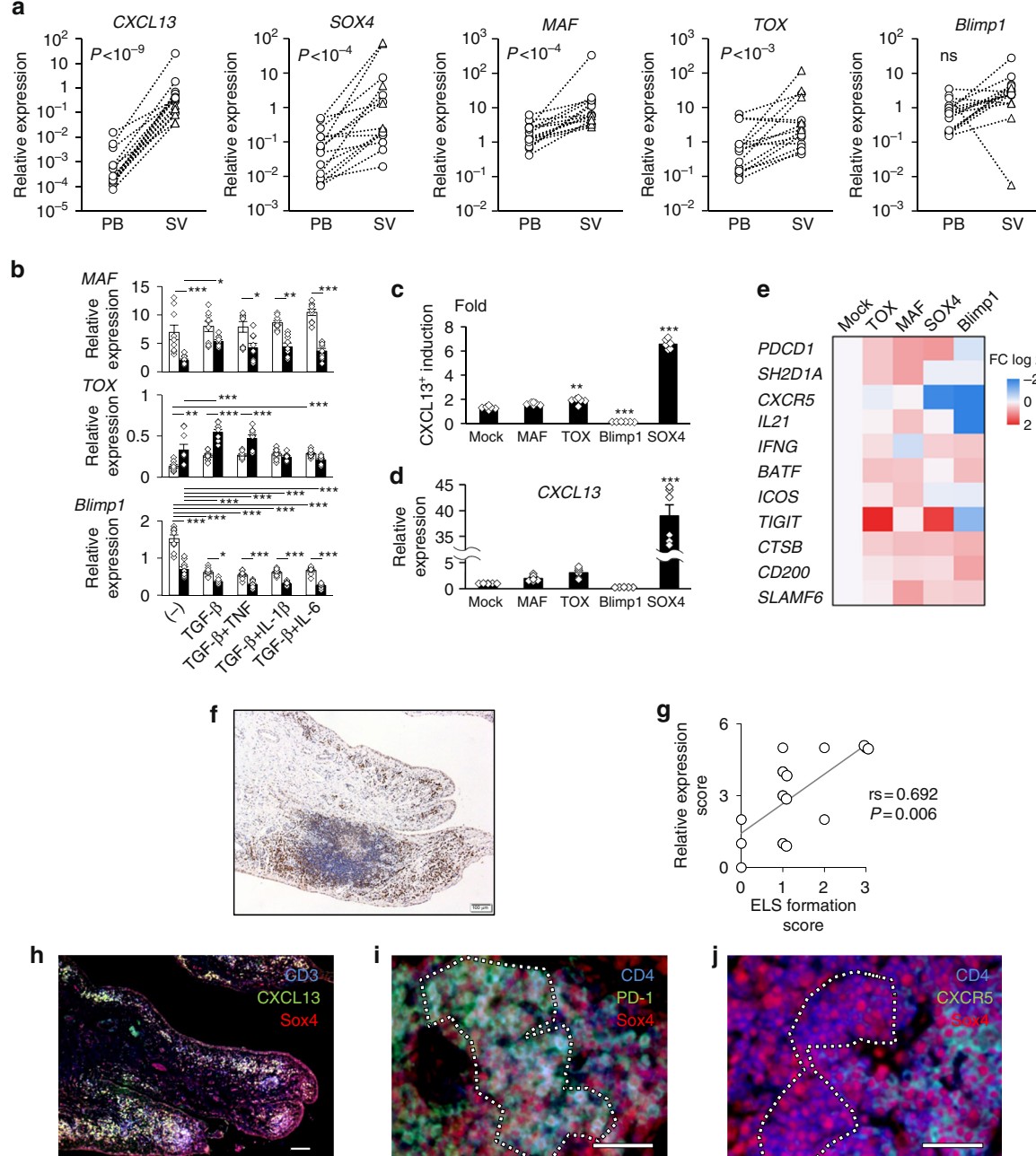

**Fig. 5** Upregulated Sox4 expression in CD4[+] T cells at local inflammatory sites correlates with ELS formation. **a** Relative expression of the indicated mRNA in paired peripheral blood (PB) and synovial (SV) CD4[+] T cells (open circle: synovial fluid, $n = 10$; open triangle: synovium, $n = 5$) of RA patients (total $n = 15$) with paired *t*-test *P*-values. **b** Expression of *MAF*, *TOX*, and *Blimp1* in CD4[+] T cells differentiated with the indicated cytokines in the presence (solid) or absence (open) of neutralizing anti-IL-2 antibody as in Fig. 2g. $n = 9$. **c, d** Human CD4[+] T cells transduced with MAF, TOX, Blimp1, or Sox4 were differentiated for 5 days. Relative induction of CXCL13-positive cells in YFP[+] cells compared with YFP[-] cells (**c**) and relative *CXCL13* expression in sorted YFP[+] cells (**d**) are shown. $n = 6$. **e** Heat map visualization of gene expressions assessed by quantitative PCR in sorted YFP[+] T cells transduced with the indicated transcription factor. Gene expression was normalized by that in mock-transduced cells in **d** and **e**. **f, g** RA synovial tissues were stained with Sox4 antibody (brown) and hematoxylin (blue). A representative image (**f**) and graphical summary of scores for ELS formation and Sox4 expression with *P*- and rs values of Spearman's rank correlation (**g**, $n = 14$) are shown. **h–j** Triple immunostaining of RA synovial tissues with the indicated antibodies. Hashed lines indicate representative PD-1[+]CD4[+] (**i**) and CXCR5[-]CD4[+] (**j**) populations. Scale bars: 100 μm in **h** and 50 μm in **i, j**. See Supplementary Figures 9 and 10 for detailed results. The data are presented as the mean ± SEM. *$P < 0.05$, **$P < 0.01$, and ***$P < 0.001$ one-way ANOVA, Tukey's test; ns, not significant

are the main source of CXCL13 among synovial T cells in RA[2,5], our findings in vitro and in vivo collectively imply that Sox4 expression in combination with the status of the intracellular signals leads to the preferential production of CXCL13 by PD-1[hi]CXCR5[-]CD4[+] T cells in inflamed joints of RA. Consistently,

triple immunostainings of CD4, PD-1, and Sox4 or of CD4, CXCR5, and Sox4 showed Sox4 expression in CD4[+] T cells positive for PD-1 and negative for CXCR5 (Fig. 5i, j and Supplementary Fig. 10). Thus, Sox4 is involved in CXCL13

production by PD-1[hi]CXCR5[−]CD4[+] T cells at inflammatory sites and also possibly in the resulting ELS formation.

## Discussion

In this study, we identified Sox4 as a transcription factor that facilitates the development of human CXCL13-producing CD4[+] T cells at inflammatory sites. Sox4 expression was upregulated in CD4[+] T cells in inflamed joints of RA and associated with ELS formation in RA synovium. Two crucial factors for the regulation of Sox4 expression in CD4[+] T cells were identified: TGF-β signaling and IL-2 levels. TGF-β is involved in the differentiation of several CD4[+] T-cell subsets, including Th17 and iTreg cells[20,26,32,33]. Interestingly, TGF-β separately regulates the expression and activity of RORγt, the master regulator of Th17 cells; TGF-β upregulates RORγt expression but downregulates the RORγt production of IL-17[26]. IL-2 is another crucial factor that regulates Th subsets. The presence of IL-2 is essential for the survival and proliferation of Treg cells[34], whereas the limiting of IL-2 is also important for promoting Tfh cell development[35]. The availability of IL-2 is attributed to the amount of IL-2 production by resident and infiltrating cells[17] and of IL-2 consumption by DC and Treg cells[18,19]. In addition, both TGF-β signaling and the suppressed IL-2 promote Sox4 expression and its promotion of CXCL13 production. In other words, CXCL13 production by human CD4[+] T cells might be finely regulated by the Sox4 expression and the surrounding environment. Moreover, CXCL13 production by Sox4-expressing CD4[+] T cells not only in TGF-β-positive, IL-2-limiting conditions but also in Th1-polarizing conditions might imply the possible involvement of Sox4 in ELS formation via CXCL13 in Th1-dominant conditions, such as tumor microenvironments, in addition to the locally inflamed sites of RA.

The transcription factors MAF and TOX are upregulated in PD1[hi]CXCR5[−]CD4[+] T cells, which constitute a distinct CD4[+] T-cell subset related to the pathogenesis of RA[5]. In vitro culture showed that the expression of these transcription factors is regulated by inflammatory conditions, but in a manner different from that of Sox4, implying that the expression of transcription factors and the phenotypes of CD4[+] T cells might be regulated by the surrounding inflammatory environment. Indeed, Sox4 (upregulated by IL-2-limiting) intensively induced CXCL13 production, whereas MAF (downregulated by IL-2-limiting) was involved in the expression of genes related to B-helper activity. These effects imply a division of roles in their regulation of human CD4[+] T cells: Sox4 contributes to PD1[hi]CXCR5[−]CD4[+] T cells in ELS formation via CXCL13 production, whereas MAF does so in B-helper function. The transcription repressor Blimp1 also intensively modulates the expression of signature genes of PD1[hi]CXCR5[−]CD4[+] T cells, including CXCL13. Although Blimp1 expression did not change significantly in our clinical samples, regulation of this transcription repressor might be important for the pathogenesis of some inflammatory diseases, as shown in our in vitro experiments.

In several inflammatory diseases, CXCL13 expression at local inflammatory sites is clinically associated with disease activities and prognosis[3,6,8,36–38], implying the strong involvement of CXCL13-dependent ELS formation in the pathogenesis. ELSs, which are discrete clusters of T cells, B cells, macrophages, and dendritic cells, support several types of immune responses[6], including an enhanced class-switch and intensive autoantibody production in RA synovium[10,39], and also better outcomes of patients with human breast cancer[3]. Furthermore, PD-1[hi]CXCR5[−]CD4[+] T cells are a source of CXCL13 at the inflammatory site in these diseases[2,3,5]. Our in vitro and in vivo results of T cells at the inflammatory sites imply that CD4[+] T cells infiltrating these sites

give rise to Sox4[+]CXCL13[+]CD4[+] T cells in response to the local inflammatory environment and antigen experience to trigger ELS formation. Unlike human CD4[+] T cells, however, mouse CD4[+] T cells do not produce CXCL13[28]. Indeed, the upregulation of Sox4 failed to induce CXCL13 in mouse CD4[+] T cells despite the abundant expression of Sox4 protein. These findings indicate that in vitro and in vivo analysis of the Sox4/CXCL13 axis using mouse models may be misleading. Further understanding the role of Sox4 and other transcription factors relating to human CD4[+] T cells may expand our knowledge of human immune responses at inflammatory sites and provide new insights into human immunology and inflammatory diseases.

## Methods

**Preparation of specimens.** Ethical approval for this study was granted by the ethics committee of Kyoto University Graduate School and Faculty of Medicine. Written informed consent was obtained from all study participants. Peripheral blood mononuclear cells (PBMCs) from healthy volunteer were collected using Lymphocyte Separation Solution 1.077 (Nacalai Tesque). Blood CD4[+] T cells were isolated with CD4 T Cell isolation kit (Miltenyi Biotec). Naive blood CD4[+] T cells were purified with naive CD4 T Cell isolation kit II (Miltenyi Biotec) through the magnetic column twice as a fraction negative for CD8, CD14, CD15, CD16, CD19, CD25, CD34, CD36, CD45RO, CD56, CD123, TCRγ/δ, HLA-DR, or CD235a. The purity of CD3[+]CD4[+]CD45RA[+] cells in the sorted cells was more than 98%.

Joint specimens were obtained as excess material during joint surgery or an outpatient clinic from patients with RA who fulfilled the American College of Rheumatology 1987 criteria[40] or 2010 ACR/EULAR classification criteria[41]. The characteristics of the RA patients are shown in Supplementary Table 3. Synovial tissues were minced and digested with 2.5 mg/ml collagenase (Roche) at 37 °C for 1.5 h and then analyzed. Blood samples were collected together with clinical blood tests. PBMCs or synovial mononuclear cells were collected using Lymphocyte Separation Solution 1.077 (Nacalai Tesque). CD4[+] T cells were sorted with Aria II (BD Biosciences) as a population of CD3[+]CD4[+] T cells.

Mouse CD4[+] T cells were isolated with CD4 microbeads (L3T4, Miltenyi Biotec) from pooled splenocytes and lymph node cells of 6–8 weeks old female BALB/cCrSlc (Japan SLC, Inc). All mice were maintained in a specific pathogen-free condition, and all animal studies were conducted in accordance with the principles of the Kyoto University Committee of Animal Resources, which are based on the International Guiding Principles for Biomedical Research Involving Animals.

**Cell culture.** Human T cells were differentiated for 5 days in a humidified, 5% CO$_2$ incubator at 37 °C with IMDM (Thermo Fisher) supplemented with 10% fetal bovine serum (Thermo Fisher), 100 units/ml penicillin and streptomycin (Thermo Fisher) under stimulation with plate-bound 5 μg/ml anti-CD3 (OKT3, Thermo Fisher), and 10 μg/ml anti-CD28 (CD28.2, Thermo Fisher) antibodies in the presence of 2 ng/ml TGF-β1 (Cell Signaling Technology) unless otherwise described. For Th1 differentiation, cells were stimulated with 5 μg/ml anti-CD3, 10 μg/ml anti-CD28, and 5 μg/ml neutralizing anti-IL-4 antibodies (BioLegend) and 10 ng/ml IL-12 (Peprotech); for Th2 differentiation, 2.5 μg/ml anti-CD3, 5 μg/ml anti-CD28, and 5 μg/ml neutralizing anti-IFN-γ antibodies (BioLegend) and 20 ng/ml IL-4 (Peprotech). For the staining of IL-4 and IFN-γ, cells were cultured for 5 h with 4 μM monensin, 10 ng/ml PMA (Nacalai Tesque), and 1 μM ionomycin (Nacalai Tesque). Recombinant tumor necrosis factor and IL-1β were obtained from Miltenyi. For the analysis of Sox4 expression 24 h after stimulation, human naive CD4[+] T cells were stimulated with anti-CD3/CD28 antibodies in serum-free X-VIVO™ 15 medium (LONZA) in the presence of the indicated cytokines (each 10 ng/ml), antibodies (each 10 μg/ml), and signal inhibitors (SB431542 (Stemgen) or SIS3 (Cayman)). Mouse CD4[+] T cells were differentiated for 5 days under stimulation with plate-bound 1 μg/ml anti-CD3 (145−2C11, BioLegend) and 5 μg/ml anti-CD28 (37.51, BioLegend) in complete RPMI-1640 (Nacalai Tesque) supplemented with 10% fetal bovine serum, non-essential amino acid (Thermo Fisher), penicillin and streptomycin (Thermo Fisher), and 2-mercaptoethanol (Thermo Fisher).

**Microarray analysis.** For the comprehensive mRNA analysis of CXCL13-producing CD4[+] T cells, human blood CD4[+] T cells labeled with CellTrace™ Violet (Thermo Fisher) were stimulated with 10 μg/ml anti-CD3 and 10 μg/ml anti-CD28 antibodies in the presence of 10 μg/ml neutralizing anti-IL-2 antibody (R&D Systems) with TGF-β1 plus IL-1β, TGF-β1 plus IL-6 (Wako), or IL-12 (each 10 ng/ml) plus 10 μg/ml neutralizing anti-IL4 antibody for 5 days in serum-containing IMDM. Cells divided more than once were sorted with MoFlo Astrios (Beckman Coulter). Dead cells were excluded with 7-amino-actinomycin D (Sigma Aldrich). mRNA was extracted using RNeasy Plus mini kit (Qiagen). Comprehensive mRNA expression was analyzed with Human Genome U133 Plus 2.0 Array according to the manufacturer's protocol. Data were assessed by GeneSpring version 12.2 (Agilent). We obtained microarray data of human naive CD4[+] T cells (CD45RA[+]CD25[−]CD4[+], Fraction VI) and human FoxP3[+] Treg cells (CD45RA[−]CD25[hi]CD4

+, Fraction II) registered by Miyara et al.[21] in the GEO database (accession number GSE15659). Genes upregulated in Fraction II more than fourfold compared with Fraction VI were excluded from the candidates as Treg-related genes.

**Lentivirus production and transduction**. The complementary DNAs encoding human *SOX4* (NM_003107.2), *MAF* (NM_005360.4), *TOX* (NM_014729.2), *PRDM1* (NM_001198.3, coding Blimp1), or mouse *Sox4* (NM_009238.2) were cloned by PCR and inserted into the multi-cloning sites of CSII-EF-MCS-IRES2-Venus (provided by Dr. H. Miyoshi, RIKEN BioResource Center), which expresses a variant of YFP Venus, which was developed by Atsushi Miyawaki. Truncation of the N-terminus domain (2aa–56aa), HMG (57aa–133aa), GRR (134aa–227aa), SRR (228aa–397aa), or TAD (423aa–474aa) of human Sox4 was performed with KOD-Plus-Mutagenesis KIT (TOYOBO). For the short hairpin RNA (shRNA) lentivirus, synthesized oligonucleotides were inserted into pENTR/U6 vector and subsequently transferred to the lentiviral destination vector CS-RfA-EG (provided by Dr. H. Miyoshi, RIKEN BioResource Center), which expresses green fluorescent protein (GFP) with a Gateway destination cassette, by performing an LR clonase reaction (Life Sciences). The inserted sequences for shRNA were as follows: for human Sox4, 5′-GGA GGA ACT CCT GCC ATT CTT CTC GAG AAT AAT GGC AGG AGT TCC TCC TTT TTT-3′; and negative control, 5′-ATC CGC GCG ATA GTA CGT ATT CTC GAG AAT ACG TAC TAT CGC GCG GAT TTT TTT-3′. HEK293T cells were transiently transfected with the expression vector, a packaging plasmid psPAX2, and an envelope plasmid pMD2.G (gifts from Didier Trono; Addgene plasmid #12259 and # 12260). Lentiviral supernatants were collected 72 h after transfection, concentrated by ultracentrifugation at $36,000 \times g$ for 2 h, washed with Hank's Balanced Salt Solution (Wako), and ultracentrifuged again to prevent the carryover of culture medium. Human naive CD4$^+$ T cells were stimulated with anti-CD3/28 antibodies for 24 h without cytokines, transduced with lentiviral supernatant at a multiplicity of infection of 10–50 by 90 min centrifugation at $3200 \times g$ and 32 °C. TGF-β1 at 2 ng/ml was added just after the transduction, followed by analysis on day 5. Mouse CD4$^+$ T cells were stimulated with anti-CD3/28 antibodies for 48 h, transduced with lentiviral supernatant at a multiplicity of infection of 10–50 in the presence of 6 μg/ml polybrene without spin infection, moved to a new culture plate with fresh complete RPMI medium 24 h after transduction, and collected for analysis on day 5.

**Quantitative PCR**. mRNA was extracted using RNAeasy mini or micro kit (Qiagen). cDNA synthesis was performed using SuperPrep® RT Kit for qPCR (TOYOBO). qRT-PCR was performed using THUNDERBIRD™ SYBR® qPCR Mix (TOYOBO) on StepOne Plus (Thermo Fisher) with the following primers:

*hCXCL13* (5′-TCTCTGCTTCTCATGCTGCT-3′ and 5′-TCAAGCTTGTGTAATAGACCTCCA-3′),

*h18srRNA* (5′-AACTTTCGATGGTAGTCGCCG-3′ and 5′-CCTTGGATGTGGTAGCCGTTT-3′),

*hSOX4* (5′-AAGATCATGGAGCAGTCGCC-3′ and 5′-CGCCTCTCGAATGAAAGGGA-3′),

*hPDCD1* (5′- CTCCAGGCATGCAGATCC-3′ and 5′-GGCCTGTCTGGGGAGTCTA-3′),

*hSH2D1A* (5′-AGGCGTGTACTGCCTATGTG-3′ and 5′-GTACCCCAGGTGCTGTCTCA-3′),

*hCXCR5* (5′-GCCATGAACTACCCGCTAAC-3′ and 5′-TCTGTCCAGTTCCCAGAACA-3′),

*hIL21* (5′-AGGAAACCACCTTCCACAAA-3′ and 5′-GAATCACATGAAGGGCATGTT-3′)[3],

*hIFNG* (5′-GCATCGTTTTGGGTTCTCTTG-3′ and 5′-AGTTCCATTATCCGCTACATCTG-3′)[3],

*hBATF* (5′-ACACAGAAGGCCGACACC-3′ and 5′-CTTGATCTCCTTGCGTAGAGC-3′),

*hICOS* (5′-GGATGCATACTTATTTGTTGGCTTA-3′ and 5′-TGTATTCACCGTTAGGGTCGT-3′),

*hTIGIT* (5′-GCTGGTGTCTCCTCCTGATCT-3′ and 5′-TGTGCCTGTCATCATTCCTG-3′),

*hCTSB* (5′-GGCTATGTGGTACCTTCCTGG-3′ and 5′-GCTTCAGGTCCTCGGTAAACA-3′),

*hCD200* (5′-AGGATGGAGAGGCTGGTGA-3′ and 5′-ACCACTGCTGCCATGACC-3′),

*hSLAMF6* (5′-TCCAATCGCTCCTGTTTGTCT-3′ and 5′-AAGAGTTACTGACTCCCCCAGA-3′),

*hTOX* (5′-CCTGCCTGGACCCCTACTAT-3′ and 5′-CTGGCTGGCACATAGTCCTG-3′),

*hBCL6* (5′-GTTTCCGGCACCTTCAGACT-3′ and 5′-CTGGCTTTTGTGACGGAAAT-3′)[3],

*hBlimp1* (5′-AACTTCTTGTGTGGTATTGTCGG-3′ and 5′-TCTCAGTGCTCGGTTGCTTT-3′)[3],

*hMAF* (5′-CACCCTGCTCGAGTTTGTG-3′ and 5′-CATGAGCCAGACACCCATT-3′),

*mSox4* (5′-ACAGCGACAAGATTCCGTTC-3′ and 5′-GTCAGCCATGTGCTTGAGG-3′),

*mCxcl13* (5′-CATAGATCGGATTCAAGTTACGCC-3′ and 5′-TCTTGGTCCAGATCACAACTTCA -3′), and

*mGapdh* (5′-GTGTTCCTACCCCCAATGTGT-3′ and 5′-ATTGTCATACCAGGAAATGAGCTT-3′).

The expressions of human and mouse mRNA were normalized by that of *h18srRNA* and *mGapdh*, respectively.

**Immunoblot analysis**. For immunoblotting, 10–20 μg of cell lysate was subjected to 10% polyacrylamide gel (ATTO) and transferred onto a polyvinylidene difluoride membrane (Millipore)[42]. After blocking with 5% skim milk, the membrane was probed with anti-human/mouse Sox4 rabbit polyclonal antibody (AB5803, Merc Milipore; 1:200) followed by incubation with horseradish protein (HRP)-conjugated anti-rabbit IgG antibody (7074, CST, 1:3000), or the membrane was probed with HRP-conjugated anti-human/mouse β-actin antibody (AC-15, Sigma Aldrich; 1:10,000). The membrane was visualized using ECL Prime Western Blotting Detection Reagent (GE Healthcare).

**Flow cytometry**. For intracellular staining, cells were cultured for 5 h with 4 μM monensin (Sigma-Aldrich), fixed, and stained with eBioscience™ Intracellular Fixation & Permeabilization Buffer (Thermo Fisher) and antibodies for intracellular molecules. Cells transfected with YFP- or GFP-expressing lentiviral particles were fixed with 4% paraformaldehyde (Nacalai Tesque) instead of eBioscience™ Intracellular Fixation & Permeabilization Buffer to avoid the bleaching of YFP or GFP. Data were acquired with a FACS Canto II flow cytometer (BD Biosciences) and were analyzed with FlowJo 10.0.8 (Tree Star). Fixable Viability Dye eFluor 780 or eFluor 506 (Thermo Fisher) was used to exclude dead cells. The border between positive and negative populations was determined according to the staining with isotype controls. PE-conjugated anti-IL-4 (8D4-8), PE/Cy7-conjugated anti-PD-1 (EH12.2H7), APC/CY7-conjugated anti-CD4 (RPA-T4), and BV421-conjugated anti-CD3 (UCHT1) were obtained from Biolegend. Fluorescein isothiocyanate (FITC)-conjugated control mouse IgG1, PE-conjugated control mouse IgG1, rat IgG2a or anti-TOX (TXRX10), and eFluor660-conjugated anti-cMAF (sym1F1) were obtained from Thermo Fisher. APC-conjugated or PE-conjugated anti-CXCL13 (53610) and control mouse IgG1 were obtained from R&D Systems. FITC-conjugated anti-IFN-γ (B27) and FITC-conjugated anti-HLA-DP, DQ, and DR (CR3/43) were obtained from BD Bioscience and DAKO, respectively.

**Immunohistochemistry and immunofluorescence**. Paraffin-embedded synovial tissues of RA were sectioned, deparaffinized, and antigen-retrieved with 0.01 M citric acid of pH 6.0. The clinical background of the patients is shown in Supplementary Table 3. Immunohistochemistry was performed with Sox4 antibody (HPA029901, rabbit polyclonal, Sigma Aldrich, 1:100) and detected with EnVisionTM Detection System (DAKO). ELS formation (Score E) and the presence of Sox4-positive cells within the infiltrating cell population without ELSs (Score SN) or with ELSs (Score SE) were assessed by a semi-quantitative four-point scale (none, 0; mild, 1; moderate, 2; and severe, 3). The total Sox4 expression score corresponds with the sum of the SN and SE scores. Triple immunofluorescence staining was performed with CXCL13 antibody (BCA1, rabbit polyclonal, Thermo Fisher, 1:100), Sox4 (HPA029901, rabbit polyclonal, Sigma Aldrich, 1:30), CD3 (Clone PS1, mouse monoclonal, Leica Microsystems, 1:200), CD4 (Clone 1F6, mouse monoclonal, Leica Microsystems, 1:100), CXCR5 (Clone 51505, mouse monoclonal, R&D Systems, 1:1000), and PD-1 (Clone NAT105, mouse monoclonal, Abcam, 1:50) and detected with TSA Plus Fluorescence Kit (PerkinElmer, Inc.). Fluorescence imaging analysis was performed using the FSX100 Fluorescence Microscope (Olympus).

**Statistical analysis**. Statistical significance was determined using a two-tailed Student's *t*-test, one-way analysis of variance with Tukey's multiple comparison test, Pearson's correlation analysis, or Spearman's rank correlation as appropriate with JMP Pro 13 (JMP). *P*-values of < 0.05 were considered statistically significant. No pre-specified effect size was calculated and no statistical method was used to predetermine sample size.

## Data availability

The affymetrix data have been deposited in the GEO database under the accession number GSE117139. The data supporting the findings of this study are available upon reasonable request to the authors.

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

## Acknowledgements

We thank N. Deguchi, K. Kokuryo, A. Yoshizawa, K. Hirota, T. Kajitani, M. Hikida, T. Watanabe, and S. Narumiya for technical support and constructive discussion, and P. Karagiannis for proof reading. This work was supported by the Special Coordination Funds for Promoting Science and Technology from Ministry of Education, Culture, Sports, Science and Technology and Astellas Pharma, Inc., and by JSPS KAKENHI Grant Number JP16K15663.

## Author contributions

H. Y. contributed to the design of the project, conducting the experiments, data analysis, and writing of the manuscript. S.K. and K.D. contributed to cellular and molecular experiments. A.M.H., A.O., T.T., and H.H. contributed to the immunohistochemistry, immunofluorescence, and scoring. K.N., K.M., H.I., S.M., and J.T. contributed to the patient recruitment and sample acquisition. Astellas Pharma had no role in the study design or in the collection, analysis, or interpretation of the data, the writing of the manuscript, or the decision to submit the manuscript for publication. Publication of this article was approved by an intellectual property committee composed of representatives from Kyoto University and Astellas Pharma.

## Additional information

**Competing interests:** H.Y., T.T., and H.I. declare the following competing interests that they received research funding from Astellas. The remaining authors declare no competing interests.

