## [Peer Review File · Nature Communications]

Reviewers' comments:

Reviewer #1 (Autoimmunity, RA, ELS)(Remarks to the Author):

In this manuscript, Yoshitomi and colleagues present data claiming to identify Sox4 as the transcription factor facilitating the development of CXCL13-producing helper T cells in the inflammatory environment of the rheumatoid arthritis (RA) synovium. This work follows on the recent description of PD-1hiCXCR5-CD4+ T cells (Rao et al. 2017), which are expanded in the peripheral blood of RA patients and support the formation of ectopic lymphoid structures (ELS) in RA synovium, as previously reported by Manzo et al. 2008 and Kobayashi et al. 2013. Since the origin and differentiation/regulation of these cells is still undefined, the authors attempt to define the conditions leading to the differentiation of this cell subset and the transcription factors involved.

Though the manuscript summarises an impressive amount of work and the in vitro gain and loss of function experiments are convincing, the data presented do not fully support their conclusions.

The main reason for this is that the data presented both in terms of Sox4 expression and CXCL13 induction in peripheral blood (PB) of healthy volunteers and/or PB and/or synovial cells from RA patients is shown mainly in total CD4+ T cells, rather than in PD-1hiCXCR5-CD4+ T cells.

In addition, it is unclear whether, or in which experiments, synovial cells (defined as SV) are derived from synovial tissue or synovial fluid, see comment on Figure 5 below.

Furthermore, in the immunofluorescence experiments in RA ST, they performed triple immunostainings with anti-CD3, Sox4, and CXCL13 antibodies (Fig 5h) but they did not stain for CD4 or PD1 nor did they demonstrate these cells were negative for CXCR5 expression.

Therefore, though it is possible that the increased expression of Sox4 shown in the RA ST, resulting from generalised activation secondary to the highly inflammatory environment rich in TGF β but low in IL-2 may lead to transcriptional regulation of CXCL13 production by PD-1hiCXCR5-CD4+ T cells, no direct proof of this is provided.

Additional concerns include:

- The sample sizes and statistical approaches are not clear. Only too often the authors show representative experiments out of three, which is not always an acceptable number when analysing relatively rare cell populations, and often without showing cumulative data. As for the statistical approach, they only mention using t test but often compare multiple conditions (e.g. most graphs of figure 2, in particular 2g and 2h). Using the appropriate statistical test (e.g. ANOVA) with such small sample size the statistical significance may change;
- Gating strategies (figure 1) should be shown and better explained. They should show that by gating on CD4+ T cells, the population PD-1hiCXCR5-MHC Class II+ is the subset with the increased expression of CXCL13, after TGF- β stimulation in IL-2 limiting conditions.
- From Figure 2d (immunoblot) the TGF β upregulation of Sox4 protein expression does not appear to be as impressive as the QT-PCR data, which to certain extent calls into question the principle claim of the authors
- From Fig 5a, it is unclear whether the comparison of CXCL13, SOX4, MAF, TOX and Blimp1 in CD4+ T cells between peripheral blood (PB) and synovial (SV) CD4+ T cells refers to synovial fluid or synovial tissue. Unfortunately, this issue could not be clarified even by detail reading of the methods on line, which state: "Joint specimens were obtained as excess material during joint surgery or an outpatient clinic of RA treatment from patients with RA", one presumes that at outpatient clinics only synovial fluid and not synovial tissue is obtained ...!!!

Minor concerns:

- Figure 1 and Supplementary Figure 1 have very similar figure legends and should be modified to better reflect reference to the Figure content. Moreover in the Supplementary Figure 1, it is not clear how the percentage of PD-1+ and PD-1+CXCL13+ cells is calculated (frequency of which subset?).
- Four of the five FACS plots shown in Figure 2 are also shown in Supplementary Figure 2 (in which the FACS panels axes legends are missing), which should be avoided;
- In Figure 2b and Supplementary Figure 2 (2a) the FACS plots axes legends are missing.
- In Figure 2, the sequential labelling and topographical distribution of the panels do not follow an alphabetical logical progression e.g. Fig 2 g and f.
- In Figure 5, though the authors refer to Fig 2g for the legend of the panels in 5b, the solid and open bars should be re-described here.

There are a number of grammatical imprecision and/or misspelled words. Few examples include:

- Page 2, 2nd paragraph: "with the activities of master transcription factor BCL611-13, secrete CXCL1314,15" would read better "through the activity of the master transcription factor BCL611-13, secrete CXCL1314,15."
- Page 2, 3rd paragraph: "CXCL13-producing human CD4+ T cells16" should read "CXCL13-producing human CD4+ T cells16"
- Page 4: Sox4 and CXCL13 expression in cells differentiated for three day (Fig. 2g, h) should read "Sox4 and CXCL13 expression in cells differentiated for three days (Fig. 2g, h)"

Reviewer #2 (T cell transcription factor regulation)(Remarks to the Author):

Yoshitomi et al investigate the role of transcription factor Sox4 in the acquisition of effector functions by human CD4 T cells. They report that Sox4 RNA is preferentially expressed in cells activated in the presence of TGFb (as previously reported in mouse cells [Ref.24]) and after neutralization of IL-2 activity, a condition that also leads to the production of Cxcl13. Furthermore, enforced expression of Sox4 promotes Cxcl13 production; this effect is unique to human cells and is synergic with TGFb or anti IL-2 treatment. Additional analyses, by nature essentially descriptive, bring support to the idea that Sox4 is important for Cxcl13 production by a subset of PD1+ Cxcr5- cells in rheumatoid arthritis.

This study of effector T cells differentiation should be of interest to a broad range of scientists and clinicians. The data is of high quality and the manuscript succinct and well written. Additional experiments are needed to support the relevance of main conclusion that Sox4 promotes Cxcl13 expression.

Main comments

1-The up-regulation of Sox4 protein by TGFb in vitro is rather modest (Fig. 2d). Comparisons of Sox4 protein expression in transduced cells vs. endogenous levels (after TGFb and anti IL-2 treatment) are needed to assess the physiological relevance of Sox4-mediated Cxcl13 induction.

2-Conversely, does shRNA inhibition of Sox4 expression affect Cxcl13 expression?

3-In Fig. 4d, the data is inconsistent with other conclusions that the authors', namely that Sox4 does induced Cxcl13 expression is unique to human cells. First, there is no comparison of Sox4 expression levels between mouse and human cells (and proteins) Additionally, there may be qualitative differences between human and mouse proteins despite their similarity (e.g. it is possible that regions

of the mouse protein between the HMG and TAD domains would impair its ability to activate Cxcl13, even though the human GRR and SRR domain are not necessary for the activity). Perhaps the simplest way to address all such issues is to express hSox4 in mouse cells.

Other comments.

-Is the data in Fig. 5e RT-PCR, microarray?

-a better description of the analysis scheme should be provided fro Fig. 2a.

-the subtitle to the last paragraph of the results section and its initial sentence are not supported by the results. The data is correlative only, not demonstrative.

Reviewer #1 (Autoimmunity, RA, ELS)(Remarks to the Author):

In this manuscript, Yoshitomi and colleagues present data claiming to identify Sox4 as the transcription factor facilitating the development of CXCL13-producing helper T cells in the inflammatory environment of the rheumatoid arthritis (RA) synovium. This work follows on the recent description of PD-1hiCXCR5-CD4+ T cells (Rao et al. 2017), which are expanded in the peripheral blood of RA patients and support the formation of ectopic lymphoid structures (ELS) in RA synovium, as previously reported by Manzo et al. 2008 and Kobayashi et al. 2013. Since the origin and differentiation/regulation of these cells is still undefined, the authors attempt to define the conditions leading to the differentiation of this cell subset and the transcription factors involved.

Though the manuscript summarises an impressive amount of work and the in vitro gain and loss of function experiments are convincing, the data presented do not fully support their conclusions.

The main reason for this is that the data presented both in terms of Sox4 expression and CXCL13 induction in peripheral blood (PB) of healthy volunteers and/or PB and/or synovial cells from RA patients is shown mainly in total CD4+ T cells, rather than in PD-1hiCXCR5-CD4+ T cells.

In addition, it is unclear whether, or in which experiments, synovial cells (defined as SV) are derived from synovial tissue or synovial fluid, see comment on Figure 5 below. Furthermore, in the immunofluorescence experiments in RA ST, they performed triple immunostainings with anti-CD3, Sox4, and CXCL13 antibodies (Fig 5h) but they did not stain for CD4 or PD1 nor did they demonstrate these cells were negative for CXCR5 expression.

Therefore, though it is possible that the increased expression of Sox4 shown in the RA ST, resulting from generalised activation secondary to the highly inflammatory environment rich in TGFb but low in IL-2 may lead to transcriptional regulation of CXCL13 production by PD-1hiCXCR5-CD4+ T cells, no direct proof of this is provided.

As the reviewer pointed out, we demonstrated the origin of synovial cells in the revised figure 5a. (See also the comment below for figure 5a).

Considering already reported observations that PD-1^{hi}CXCR5⁻CD4⁺ T cells are main source of CXCL13 among CD3⁺ T cell (Kobayashi, 2013; Rao, 2017), we supposed it was reasonable claiming that CXCL13⁺CD3⁺ T cells of RA synovium correspond to PD-1^{hi}CXCR5⁻CD4⁺ T cells. However, as the reviewer pointed out, additional triple stainings are preferable. In Fig. 5 j, i and supplementary figure 10, we additionally showed the results of triple stainings of PD-1, CD4, and Sox4 or CXCR5, CD4, and Sox4 showing that PD-1^{hi}CD4⁺ and CXCR5⁻CD4⁺ populations express Sox4. Collectively, these findings strongly support that PD-1^{hi}CXCR5⁻CD4⁺ T cells at inflammatory sites are involved in CXCL13 production via Sox4.

#The sample sizes and statistical approaches are not clear. Only too often the authors show representative experiments out of three, which is not always an acceptable number when analysing relatively rare cell populations, and often without showing cumulative data. As for the statistical approach, they only mention using t test but often compare multiple conditions (e.g. most graphs of figure 2, in particular 2g and 2h). Using the appropriate statistical test (e.g. ANOVA) with such small sample size the statistical significance may change;

As the reviewer commented the number of sample size in the initial manuscript was obscure. According to this comment by the reviewer and the editorial policy, we plotted each dot of cumulative data and described sample numbers in corresponding figure legends. We also used one-way ANOVA with Tukey multiple comparison test and Pearson correlation analysis, in addition to t test and Spearman's rank correlation as appropriate.

#Gating strategies (figure 1) should be shown and better explained. They should show that by gating on CD4+ T cells, the population PD-1^{hi}CXCR5⁻MHC Class II⁺ is the subset with the increased expression of CXCL13, after TGF- β stimulation in IL-2 limiting conditions.

As the reviewer commented, we showed gating strategies for figure 1 in the revised supplementary figure 1. We also add data showing that the expression of CXCL13 in PD-1^{hi}CXCR5⁻MHC Class II⁺ cells is upregulated by IL-2 limiting in supplementary figure 2. For the differentiation analysis of purified naïve CD4⁺ T cells, we gated live lymphocytes with FSC and SSC instead of CD4 staining, because CD3/28 stimulation intensively downregulates CD4 expression.

#From Figure 2d (immunoblot) the TGF β upregulation of Sox4 protein expression does not appear to be as impressive as the QT-PCR data, which to certain extent calls into question the principle claim of the authors

Thank you for pointing out this. As the reviewer commented, the initial figure failed to convey the impression of original images. We adjusted the size of images to convey correct impression in figure 2d.

#From Fig 5a, it is unclear whether the comparison of CXCL13, SOX4, MAF, TOX and Blimp1 in CD4⁺ T cells between peripheral blood (PB) and synovial (SV) CD4⁺ T cells refers to synovial fluid or synovial tissue. Unfortunately, this issue could not be clarified even by detail reading of the methods on line, which state: "Joint specimens were obtained as excess material during joint surgery or an outpatient clinic of RA treatment from patients with RA", one presumes that at outpatient clinics only synovial fluid and not synovial tissue is obtained ...!!!

As the reviewer pointed out, we need to clarify whether each synovial CD4 sample was synovial fluid or synovial tissues. In the revised figure 5a, we analyzed 10 synovial fluid samples (open circles) and 5 synovial tissue samples (open triangles). (We also increased sample numbers during this revision.) As the reviewer may know it, recent improvements of RA treatment make it difficult to obtain synovial fluid samples with inflammation containing lymphocytes enough for investigation at joint surgeries. We were able to obtain both synovial fluid CD4⁺ T cells and synovial tissue CD4⁺ T cells from one patient. The data of PB CD4⁺ T cells of the patient was used in both comparisons for synovial fluid and synovial tissue. As the reviewer also commented, synovial fluid samples were mainly obtained from patients visiting RA outpatient clinic with flared up joints.

Minor concerns:

#Figure 1 and Supplementary Figure 1 have very similar figure legends and should be modified to better reflect reference to the Figure content. Moreover in the Supplementary Figure 1, it is not clear how the percentage of PD-1⁺ and PD-1⁺CXCL13⁺ cells is calculated (frequency of which subset?).

Thank you for pointing out. We modified figure legend in supplementary figure 2 (originally supplementary figure 1). Also, mixed usage of PD-1⁺ and PD-1^{hi} (in almost

same meanings we used) was confusing. We replaced “PD-1⁺” to “PD-1^{hi}” throughout the manuscript and added schematic descriptions of the population in supplementary figure 2.

#Four of the five FACS plots shown in Figure 2 are also shown in Supplementary Figure 2 (in which the FACS panels axes legends are missing), which should be avoided; In Figure 2b and Supplementary Figure 2 (2a) the FACS plots axes legends are missing.

We removed FACS plots from figure 2b and Supplementary Figure 2. Instead, we added histograms of CXCL13 in figure 2b and brief description of analysis according to request of the other reviewer in supplementary figure 3a (originally supplementary figure 2a).

#In Figure 2, the sequential labelling and topographical distribution of the panels do not follow an alphabetical logical progression e.g. Fig 2 g and f.

#In Figure 5, though the authors refer to Fig 2g for the legend of the panels in 5b, the solid and open bars should be re-described here.

Thank you for intensive reading. We fixed them.

#There are a number of grammatical imprecision and/or misspelled words. Few examples include:

- Page 2, 2nd paragraph: “with the activities of master transcription factor BCL611-13, secrete CXCL1314,15” would read better “through the activity of the master transcription factor BCL611-13, secrete CXCL1314,15.
- Page 2, 3rd paragraph: “CXCL13-producing human CD4+ T cells16” should read “CXCL13-producing human CD4+ T cells16”
- Page 4: Sox4 and CXCL13 expression in cells differentiated for three day (Fig. 2g, h) should read “Sox4 and CXCL13 expression in cells differentiated for three days (Fig. 2g, h)”

Thank you again for intensive reading. We try to fix grammatical errors.

Reviewer #2 (T cell transcription factor regulation)(Remarks to the Author):

Main comments

#1-The up-regulation of Sox4 protein by TGFb in vitro is rather modest (Fig. 2d). Comparisons of Sox4 protein expression in transduced cells vs. endogenous levels (after TGFb and anti IL-2 treatment) are needed to assess the physiological relevance of Sox4-mediated Cxcl13 induction.

2-Conversely, does shRNA inhibition of Sox4 expression affect Cxcl13 expression?

Thank you for pointing out important issues. We added immunoblottings comparing Sox4 protein expression in transduced cells vs. endogenous levels in supplementary figure 4. Considering that transduced cells were purified by sorting, the expression level of Sox4 protein in transduced cell may be in the range of physiological expression, we think.

We also added the data of Sox4-knockdown according to the comment of the reviewer. Significant downregulation of CXCL13 induction by Sox4 knockdown implies the physiological significance of Sox4 in CXCL13 induction by human CD4⁺ T cells.

#3-In Fig. 4d, the data is inconsistent with other conclusions that the authors', namely that Sox4 does induced Cxcl13 expression is unique to human cells. First, there is no comparison of Sox4 expression levels between mouse and human cells (and proteins) Additionally, there may be qualitative differences between human and mouse proteins despite their similarity (e.g. it is possible that regions of the mouse protein between the HMG and TAD domains would impair its ability to activate Cxcl13, even though the human GRR and SRR domain are not necessary for the activity). Perhaps the simplest way to address all such issues is to express hSox4 in mouse cells.

We added immunoblotting of Sox4 protein in mouse and human CD4⁺ T cell in supplementary figure 7. According to this result, expression levels of Sox4 protein in mouse CD4⁺ T cells are rather higher than that in Sox4-transduced human CD4⁺ T cells.

Furthermore, transduction of human Sox4 into mice CD4⁺ T cells also failed to induce CXCL13, implying it is not attributed to the difference of GRR or SRR between human and mice Sox4 that upregulation of mouse Sox4 failed to induce CXCL13 in mouse CD4⁺ T cells. In figure 3, we showed the function of Sox4 is strongly affected by intracellular signaling status. Probably, the difference of transcription co-factor or promoter/enhancer regions between mice and humans might be the cause of the absence of Sox4/CXCL13 axis in mouse CD4⁺ T cells. We added this speculation in the results

part (page 5).

Other comments.

#Is the data in Fig. 5e RT-PCR, microarray?

The data is of qPCR. We fixed the figure legend of Fig. 5e.

#a better description of the analysis scheme should be provided for Fig. 2a.

We added schematic description for Fig.2a in supplementary figure 3a.

#the subtitle to the last paragraph of the results section and its initial sentence are not supported by the results. The data is correlative only, not demonstrative.

We change the sentence “Sox4 expression in local T cells contributes to ELS formation in RA synovium.” into “Sox4 expression in local T cells correlates with ELS formation in RA synovium.”

REVIEWERS' COMMENTS:

Reviewer #1 (Remarks to the Author):

The manuscript has been reviewed addressing the main concerns raised by the reviewers.

Reviewer #2 (Remarks to the Author):

The authors have addressed all my comments with appropriate new experiments. In Fig. S5bc, it should be indicated what is measured on the sorted cells: mRNA (RT-PCR?), protein levels?

Reviewer #2 (Remarks to the Author):

The authors have addressed all my comments with appropriate new experiments. In Fig. S5bc, it should be indicated what is measured on the sorted cells: mRNA (RT-PCR?), protein levels?

We described what is measured on the sorted cells in the Supplementary Fig 5c and d as below.

“b,c mRNA expression of *Sox4* relative to *18srRNA* determined by quantitative PCR in sorted GFP⁺ cells (**b**, n=3), and relative induction of cells positive for CXCL13 protein in GFP⁺ cells compared with GFP⁻ cells determined by flow cytometry (**c**, n=6 from two experiments) are shown.”